# Power and Design Issues in Crossover-Based N-Of-1 Clinical Trials with Fixed Data Collection Periods

**DOI:** 10.3390/healthcare7030084

**Published:** 2019-07-02

**Authors:** Yanpin Wang, Nicholas J. Schork

**Affiliations:** 1Model Risk Management, USAA, 9257 Delaney Creek Blv, Tampa, FL 33619, USA; 2Quantitative Medicine and Systems Biology, The Translational Genomics Research Institute (TGen), an affiliate of the City of Hope National Medical Center, 445 North Fifth Street, Phoenix, AZ 85004, USA; 3The Joint City of Hope/TGen IMPACT Center (NJS), Duarte, CA 91010, USA

**Keywords:** Efficacy, linear models, interventions, human clinical studies

## Abstract

“N-of-1,” or single subject, clinical trials seek to determine if an intervention strategy is more efficacious for an individual than an alternative based on an objective, empirical, and controlled study. The design of such trials is typically rooted in a simple crossover strategy with multiple intervention response evaluation periods. The effect of serial correlation between measurements, the number of evaluation periods, the use of washout periods, heteroscedasticity (i.e., unequal variances among responses to the interventions) and intervention-associated carry-over phenomena on the power of such studies is crucially important for putting the yield and feasibility of N-of-1 trial designs into context. We evaluated the effect of these phenomena on the power of different designs for N-of-1 trials using analytical theory based on standard likelihood principles assuming an autoregressive lag 1, i.e., AR(1), serial correlation structure among the measurements as well as simulation studies. By evaluating the power to detect effects in many different settings, we show that the influence of serial correlation and heteroscedasticity on power can be substantial, but can also be mitigated to some degree through the use of appropriate multiple evaluation periods. We also show that the detection of certain types of carry-over effects can be heavily influenced by design considerations as well.

## 1. Introduction

The belief that medical interventions and treatments can be tailored to an individual patient’s biochemical, physiological, behavioral, and environmental exposure profile has received considerable recent attention. This is almost certainly attributable to the development of technologies, such as DNA sequencing, microarray, proteomic and metabolomics-based assay technologies, which are allowing the characterization of patient-specific molecular pathologies that, in order to be overcome or treated, may need nuanced interventions [1,2]. However, as exciting as these developments are, they will not necessarily result in better interventions or treatments for individual patients unless the recommended interventions and treatments are tested objectively and proven to have benefit for those patients. Proving that an intervention works uniquely for a particular patient is not trivial, as an appropriately designed, objective study must be conducted that considers the benefits of the intervention for the patient in question, possibly without the ability to draw on the response profiles among other patients, depending on how nuanced and tailored the intervention might be [2,3,4,5,6].

Clinical trials that focus on intervention responses exhibited by individual patients have been described in the literature [3,4,5,6,7,8,9]. Such single subject or ‘N-of-1′ clinical trials have been conducted in a variety of ways, and although some recent studies have leveraged very sophisticated designs and clinical outcome measures [9,10,11], many N-of-1 studies pursued to date have lacked sophisticated experimental designs, data analysis methods, and inferential procedures [9,12]. In this context, Gabler et al. [13] performed a systematic review of the literature for N-of-1 studies and found that of the 108 N-of-1 trials they identified from the literature involving a total of 2154 individuals, only 49% of those trials used statistical criteria to determine a superior intervention, whereas others used a graphical comparison (25%) or a clinical significance cutoff (20%). Approximately two-thirds of the studies provided ongoing clinical care-based intervention change information (i.e., essentially addressing questions concerning whether or not a patient continued to get better on the proven, more favorable, intervention after the trial was completed) and of those studies, only 54% of the participants in them were provided subsequent and ongoing interventions that were consistent with the results of the trial. The authors concluded, justifiably, that greater attention to study design, data collection strategies, data analysis methods, inferential procedures and integration with ongoing clinical care should be encouraged [13].

## 2. Basic Designs For N-Of-1 Trials

Although individual N-of-1 trials cannot be generalized (by definition) unless the results of a number of N-of-1 trials are aggregated [14], they can be pursued in a way that leverages design and analysis strategies associated with standard population-based trials to make them more compelling and objective, such as blinding, the use of placebo controls (where appropriate), washout periods, crossover and randomization schemes, sequential and adaptive designs, flexible data analysis methodologies, and sophisticated treatment response collection devices [7]. We have explored some of the issues surrounding the design of N-of-1 trials that consider the comparison of two interventions, e.g., an active (A) intervention and a placebo or sham (P) intervention. Such settings are comparable to the traditional ‘AB’ design of N-of-1 trials, where ‘A’ and ‘B’ denote different treatments whose effects on a patient could be evaluated and compared as part of a study which alternates between their administrations some number of times (e.g., ‘ABAB’ or ‘ABABAB’) [15]. Such studies are known as ‘crossover’ designs since they entail the administration of different drugs provided in succession. Our results could easily be extended to settings in which two active, yet different, interventions are compared. We also focused on simple crossover-based study designs in which the allocation of interventions over the trial is fixed and non-random, and for which a fixed number of response observations are collected during each intervention administration period. For example, in comparing active (A) and placebo (P) interventions, a patient could be administered the active intervention and then the placebo (or vice versa). The response observations collected during each intervention period could then be contrasted using statistical methods (in this setting, the design could be written as a simple ‘AP’ design). If multiple periods for collecting response observations during the administration of each intervention were considered, then more sophisticated designs could be exploited (e.g., APAPAPAP) and possibly include washout (denoted ‘w’) periods (e.g., AwPwAwPwAwPwAwP). In this context, we have explored the effect of different ways of providing (or administering, or ‘stacking’, in a design sense) the drugs in succession on both practical issues and the power of such studies. Basically, we wanted to explore questions such as, ‘if one had 400 measurements, would it be better to administer the drugs in 2 successive periods (i.e., ‘AP’) with 200 measurements made during each, or 4 successive periods (i.e., APAP) with 50 measurements made in each?’ and ‘what effect, if any, do carry-over effects and serial correlation between the observations have on power, and does the use of washout periods impact power?’

## 3. Practical Considerations

Obviously the choice of a design, the ultimate manner in which observations are made, and how the periods are stacked and precede each other, will depend critically on: 1. how easy it is to collect response and outcome information; 2. at what time scales responses could be collected given the response measurement devices used (e.g., a wireless heart rate monitor could collect data 24 h a day, whereas lengthy clinical evaluations may only be conducted once a week); 3. the half-life a drug (if a drug is being tested); and 4. the condition being studied (e.g., acute and life-threatening vs. chronic), etc. In addition, there are a number of other crucial statistical phenomena associated with N-of-1 clinical trials that must be accommodated in a study design and subsequent data analyses. Among them are serial correlation, heteroscedasticity (i.e., unequal variances among the response observations across the interventions), and carryover effects, in which the influence of an intervention on response has a lingering effect on response even after it is no longer being administered (i.e., drugs often take some time to be removed completely from the body even after a patient stops taking them). Of these phenomena, serial correlation between the successively collected intervention response observations over time has been studied the most [7,16]. Serial correlation can have a pronounced effect on the power of N-of-1 studies to detect differences in response to different interventions.

We focused our analytical and simulation studies on the influence of serial correlation and the manner in which two interventions and response observations are allocated or stacked over the course of an N-of-1 trial. We also considered the effect of heteroscedasticity and the power to detect carryover effects in a trial. Our results ultimately suggest that one should be sensitive to a number of issues when designing N-of-1 trials. We note that some assumptions in our models, e.g., about the number of measurements made during each treatment period, are arbitrary with respect to a time scale (e.g., as noted, 400 measures could take 1 week or a year depending on the measurement and measurement device used; a carryover effect may last 2 or 50 measurements again depending on the time scale and the strength of the serial correlation between the observations). Therefore, we believe our results, although obtained in a limited number of settings, can provide general insights into the design of studies that could be pursued in vastly different time scales.

Given this framework, the format for the rest of the paper is as follows. We first describe the analytical model we used to assess the properties of different N-of-1 designs. We then describe how we evaluated the power of different designs using both this model and simulation studies. We then apply our model and simulation studies to address three specific and related issues: 1. The influence of serial correlation and the length(s) of the periods in which a interventions are administered (in terms of the number of response measurements made during these periods) on the power of an N-of-(1) clinical trial; (2). The influence of heteroscedasticity (i.e., unequal variances) on the power of a study; and (3). The power to detect carry-over effects as a function of serial correlation strength and the length of time a patient is administered interventions. We end with some conclusions and directions for future research.

## 4. An Analytical Model For N-Of-1 Designs

### 4.1. The Basic Linear Model

In order to evaluate the influence of various factors on the power of N-of-1 designs, we considered the use of a standard linear model of the type described by Rochon, which assumes an autoregressive, lag 1 (AR(1)) serial correlation between consecutive measurements. [16] The power of this linear model can be studied analytically using the asymptotic properties of likelihood ratio statistics derived from the model, which we describe in the following. Let $y_{i}$ be the *i*th intervention response observation (e.g., a blood pressure reading, a cholesterol level, or the amount of a circulating protein) collected on a patient out of a total of *T* such observations (i.e., i=1, …, T). Further, let the entire sequence of intervention response observations be reflected in the vector $Y=\left[y_{1},y_{2},\;\ldots ,\;y_{T}\right]$. We assume further that *C* covariates, $x_{i}$, have been collected on the patient that are hypothesized to influence the $y_{i}$ responses, and that these covariates can be collectively reflected in the *T x C+1* matrix $X=\left[x_{i,1},x_{i,2},\;\ldots ,\;x_{i,C+1}\right]$.

The covariate vector includes information that can quantify an overall mean response. This can be achieved easily by having the first covariate $x_{i,1}=1$ for all *T* observations. In addition, one of the covariates (say the last covariate in the vector, the *C + 1* covariate) can be used quantify the intervention effects through the use of dummy codes. Thus, $x_{i,C+1}=1$ if the individual was on the intervention of interest (e.g., the active compound, denoted ‘A’) and 0 if the individual was on the alternative intervention (e.g., placebo, denoted ‘P’). For example, if a total *T = 10* response observations were collected on the patient and the first 5 were collected while the patient was on the active compound and the last 5 were collected while the patient was on placebo, then column *C+1* of *X* would be: [1,1,1,1,1,0,0,0,0,0]′.

We assume that the response observations enter into a linear relationship with the covariates, which can be expressed in the simple form:(1)$$y_{i}=X_{i}B+e_{i}$$
where $X_{i}$ is the *i*th row of the covariate matrix *X*, $B=\left[b_{1},b_{2},\;\ldots ,\;b_{C+1}\right]$ is a vector of regression coefficients and $e_{i}$ is an error term. We further assume that, since all the observations have been collected over time on a single individual, the error terms exhibit serial correlation, which we assume follows an autoregressive sequence of order 1 (i.e., an AR(1) process):(2)$$e_{i}=\rho e_{i-1}+\epsilon_{i}$$where ρ is a parameter governing the strength of the serial correlation in the AR(1) process and $\epsilon_{i}$ is an error term assumed to be distributed as a normal variate *N*(*0*,$\sigma^{2}$). Given the AR(1) assumption in Equation (2), the *T × T* covariance matrix for the response observations can be written as:(3)$$\Omega \left(Y\right)=\frac{\sigma^{2}}{1-\rho^{2}}\left[\begin{array}{cc} \begin{array}{cc} 1 & \rho^{1}\\ \rho^{1} & 1 \end{array} & \begin{array}{cc} \cdots & \rho^{T-1}\\ \cdots & \rho^{T-2} \end{array}\\ \begin{array}{cc} \vdots & \vdots \\ \rho^{T-1} & \rho^{T-2} \end{array} & \begin{array}{cc} \ddots & \vdots \\ \cdots & 1 \end{array} \end{array}\right]$$
here $\sigma^{2}$ is the variance of the response observations. Note that this model assumes equal variances among the observations collected for the different interventions, an assumption that may not be valid. Unequal variances can be accounted for by writing the covariance matrix as a block diagonal matrix with unique variances for the blocks corresponding to periods during which the sequence of response observations was collected while the patient was on a specific intervention [16].

We further assume that the response observation vector follows a multivariate normal distribution, with corresponding log-likelihood function for the parameters $\left\{B,\sigma^{2},\rho \right\}$ governing the model:(4)$$L\left(B,\sigma^{2},\rho |Y,X\right)=-\frac{T}{2}\ln\left(2\pi \right)-\frac{1}{2}\ln\left|\Omega \right|-\frac{1}{2}{\left(Y-XB\right)}^{\prime }\Omega^{-1}\left(Y-XB\right).$$
It this model is evaluated with actual data, the parameters can be estimated by maximizing the likelihood provided in Equation (4) [16].

### 4.2. Evaluating Power 

In order to evaluate the power of different study designs assuming, e.g., different numbers of intervention periods for a fixed number, *T*, of observations, different serial correlation strengths, the use of washout periods between intervention periods, etc. we exploited the asymptotic behavior of likelihood ratio tests derived from the model described above. Thus, for example, to test the hypothesis of no intervention effect: $H_{0}:\;b_{C+1}=0$, one can maximize the log-likelihood in Equation (4) with $b_{C+1}=0$ and then maximize the log-likelihood assuming $b_{C+1}\neq 0$. The statistic, *LR_t_* = −*2* times the difference in log-likelihoods, follows a $\chi^{2}$ distribution with 1 degree of freedom under the null hypothesis. Under the alternative hypothesis, *LR_t_* follows a non-central $\chi_{nc}^{2}$ distribution with the non-centrality parameter, λ, calculated as a function of the difference in the values of parameters assumed under the null, $\beta_{0}$, and alterative, $\beta_{1}$, hypotheses, and the inverse of the information matrix, I, for the parameters evaluated under the null hypothesis:(5)$$\lambda =T\cdot {\left(\beta_{0}-\beta_{1}\right)}^{\prime }I^{-1}\left(\beta_{0}-\beta_{1}\right).$$

The information matrix for the model in Equation (4) is a function of the assumed parameter values in *B* and the covariance matrix Ω [16,17]. By integrating over the non-central $\chi_{nc}^{2}$ distribution with the parameter values fixed to assumed values under the null and alternative hypotheses, a specified type I error rate (in our studies, set to 0.05), and the number of observations, *T* (assumed to be 400 in our analyses), the power to detect, e.g., a treatment effect, or other phenomena specified in the parameterizations under a null and alternative hypotheses, can be computed [17].

## 5. Simulation Studies

In order to validate our analytical methods and also explore some key assumptions in the designs we considered, we performed simulation studies and compared the results with those obtained from the analytical model and calculations. We simulated N-of-1 trial setting data using the ‘arima.sim’ module in the R data analysis package [10]. We fit the model in Equation (1) to the simulated data and computed relevant test statistics using the generalized linear model module in R assuming an AR(1) covariance structure among the observations. For the different study settings, we estimated power based on 1000 simulations. As with the analytical studies, we assumed a type 1 error of 0.05 and 400 total observations. Our simulation studies focused on settings in which analytical power calculations were performed, so the same parameters that were varied in the analytical models were varied in the simulation studies.

## 6. Design Scenarios Considered

Our primary focus was on investigating the effect of breaking up the times during which a patient is on each of two treatments (e.g., active (A) vs. placebo (P)) using the analytical model. As noted, we assumed *T* = 400 total response observations and considered different ways in which the periods during which the patient was provided the two interventions were stacked. For example, we considered a design in which the first 200 observations were collected while the patient was on intervention A and the second 200 observations while on intervention P. To ease summarizing our results, we denote this design as a ‘1p x 2i’ design such that only one period (‘p’) of response collections for each of two interventions (‘i’) was considered). We use this notation in order not to confuse the reader with, e.g., 2 × 2 or 2 × 4 factorial designs common in the statistics literature. We also considered designs in which the periods during which the patient was on intervention A and P were stacked and this perforce affected the number of measures collection during each intervention or treatment period; e.g., a patient was on intervention A for the first 100 observations, intervention P for the second 100 observations, then back to drug A for the next 100 observations and then once again back to intervention P for the last 100 observations (a ‘2p x 2i’ design).

We ultimately considered designs with T = 400 observations within 1p × 2i, 2p × 2i, 4p × 2i, 8p × 2i, 10p × 2i, 20p × 2i and 40p × 2i ‘period stacking’ schemes. We further assumed a type 1 error rate of 0.05 in our studies and an initial difference in mean responses between Active and Placebo interventions of 0.3 standard deviation units. We also initially assumed a constant variance across the interventions but different serial correlation strengths between successive response observations. We only considered designs in which the interventions were alternated (e.g., APAPAPAP) and did not consider random intervention assignment designs (e.g., AAPAPPAPAP). Finally, we considered settings in which a washout period was assumed between interventions, effectively breaking up the serial correlation between the last observation in a sequence of observations while the patient is on one intervention and the first observation while the patient is switched to the other intervention.

To accommodate a washout period in the model associated with Equations (1)–(5), we included hypothetical response observations between the administrations of each intervention in the total sequence of response observations. The number of observations in the washout period (denoted ‘w’) equaled the number of observations in each intervention period (e.g., for a ‘1p × 2i’ design with *T* = 400, 200 observations were assumed during a single washout period: the design would thus be ‘AwP’ with 200 observations during the active (A), washout (w), and placebo (P) intervention administration periods); and for a ‘2p × 2i’ design with *T* = 400, 100 observations were assumed for 3 washout periods: AwPwAwP; etc.). We assumed that no measurements collected during the washout periods were used in the comparisons between the two treatments (i.e., 400 measurements associated with the non-washout periods were used to assess the difference between A’s and P’s effects on the patient’s response). Note that we assumed a constant treatment response, not a graded one while the intervention was taking effect (which would have been more realistic), and no carry-over or intervention order effects for many of the analyses.

## 7. Results

### 7.1. The Influence of Serial Correlation and Intervention Period Stacking

We considered the influence of the strength of the serial correlation parameter, ρ, on the power of designs assuming different numbers of periods. Table 1 summarizes the influence of the serial correlation strength between successive response observations on the power to detect a mean treatment difference of 0.3 standard deviations for different numbers of intervention periods and the response observations allocated to those periods. It is clear that the stronger the serial correlation between the observations, the greater the reduction in power to detect an intervention effect. However, this can be mitigated to some degree by breaking up the periods during which the patient is on each intervention, effectively making more frequent yet smaller intervention periods or response collection sequences (e.g., contrast design ‘1p × 2i’ with design ‘40p × 2i’).

Table 2 provides the results of simulation studies used to investigate the power of some of the designs considered in Table 1. In addition, Table 2 also contrasts the results of the analytical studies with the simulation studies, and shows very clearly good agreement between them. Importantly, the simulation studies also considered a setting in which no effect of the intervention is assumed, allowing the type 1 error rate of the model to be estimated. Since we assumed a type 1 error rate of 0.05 for the analytical studies, we can see that the simulation studies support the assumption that the test appropriately preserves the expected type 1 error rate of 0.05 (first 3 rows of Table 1).

We emphasize that, despite our results on the effect of the serial correlation on power, whether, given a particular dug effect or measurement device, more frequent alternations from one intervention to another is practical, unlikely to be influenced by carryover effects, or consistent with the time it takes for an intervention to induce an effect, is an open question. It is also noteworthy that the use of washout periods—when no carryover effects are present—reduces power slightly. This is likely the case because although positive correlations between measures taken during the same intervention period reduce the effective number of observations taken during that period, the positive correlation between observations taken while the patient is on different interventions increases the power to detect a difference. Essentially, the variance of the difference in two variables (e.g., responses to two different interventions) is a function of the sum of the variances associated with each variable (i.e., variance of the response while on a specific intervention) minus the covariance between the two variables. Thus, the stronger the positive covariance (correlation) between the two factors, the smaller the variance of their difference, leading to power increases (see, e.g., the work by Schork and colleagues [11,12].

Figure 1 depicts the differences in average estimated serial correlations based on simulation studies assuming an actual serial correlation of 0.5 between the observations, 3 different designs (1p × 2i = 2 total periods; 4p × 2i = 8 total periods; and 10p × 2i = 20 total periods) assuming the use of no washouts as well as the use of washouts. It is clear from Figure 1 that the serial correlation across the observations used in the analysis (i.e., those made during the administration of either of the 2 interventions) is reduced as more periods are used since there is then a greater number of washout periods. Thus, in the absence of carryover effects, there is an interesting balance between increasing power to detect a difference between two interventions by increasing the effective number of response observations taken while a patient is on a specific intervention (via breaking up the times the patient is on and off that intervention), and decreasing the power to detect a difference by reducing the positive correlation between response observations taken while the patient is on different interventions through the use of washout periods. Additional simulation studies that we pursued were also consistent with this phenomenon (data not shown).

Figure 2 depicts the influence of the effect size on power for different serial correlation strengths. A 4p × 2i design with *T* = 400 was assumed with no washout periods. Figure 2 clearly shows that effect size has a dramatic influence on power even in the presence of strong serial correlation among the response observations. This suggests for measurements that are collected frequently (e.g., every minute), such that there is likely to be strong serial correlation between them, can be used to detect pronounced effects of an intervention. Table 3 provides the results of simulation studies examining the relationship between effect sizes (again measured as the number of standard deviation units between the means of the measurements associated with each intervention) and the serial correlation between measurements as a function of different period stacking settings and confirms the negative effect on power that strong serial correlation between the observations has.

### 7.2. The Influence of Heteroskedasticity (i.e., Unequal Response Variances)

To model the influence of unequal variances associated with response observations made during the administration of the different interventions, we considered two variances, $\sigma_{A}^{2}$, and $\sigma_{P}^{2}$, for the observations collected under the administration of, e.g., an active (A) and placebo (P) intervention. We incorporated these variances into the calculation of the covariance matrix in Equation (3) using the blocking strategy discussed by Rochon [16] and discussed in the ‘An Analytical Model For N-of-1 Design’ section. Although many different scenarios were evaluated, Figure 3 depicts some basic results assuming a 4p × 2i design with *T* = 400 (i.e., 50 observations per intervention) with no washout periods. A variance twice as big for one intervention relative to the other was assumed for these calculations. Power is depicted as a function of effect size calculated as the absolute difference in means divided by the pooled standard deviation of the response observations obtained for the two different intervention periods. It can be seen from Figure 3 that the power to detect an effect is influenced not just by the serial correlation strength and effect size, but also by unequal variances. Thus, for small effect sizes, unequal variances for the two treatments, which could affect the serial correlation between observations in pronounced ways, increases power in the presence of a weak overall serial correlation, but this effect is opposite for large effect sizes and not as pronounced in the presence of a strong serial correlation between the observations.

### 7.3. Detecting Carryover Effects

As noted, carryover effects are likely to arise in N-of-1 clinical trials given that it takes some time for, e.g., most pharmaceutical compounds to dissipate, be eliminated from the body, otherwise no longer leave their mark on bodily function in some way. Although there are many ways of modeling and testing for carryover effects in crossover trials, we explored one simple method analytically. We assumed that one could contrast the mean response observed during an initial intervention administration period with the mean response obtained from the combined subsequent intervention administration periods. If no carryover effects occurred, then the mean response from the initial intervention administration period should equal the mean response across the subsequent intervention administration periods. We assumed different designs, as with the discussion surrounding the results provided in Table 1, with *T* = 400, with no washout periods. The number of observations in the initial intervention administration period was assumed to be equal to each subsequent intervention administration periods. Thus, if a 4p × 2i design was assumed, there would be 50 observations collected during the initial administration period of say, the active intervention (A), and then 50 during each of 3 subsequent intervention administration periods of the active intervention (e.g., comparing average responses to A in first AP period vs. the remaining APAPAP periods). This would total 150 observations across all the subsequent intervention administration periods. Note that this formulation assumes that the comparator intervention, here assumed to be a placebo (P), induces a carryover effect on the active intervention (A), but one could easily assume the opposite or just assume A and P correspond to two active interventions. We assumed an effect size of 0.3 standard deviation units between the initial intervention administration period and the subsequent intervention administration periods. Table 4 provides some results and suggests that the power to detect a carryover effect is weak in the settings we chose, but is clearly strongly influenced by the strength of the serial correlation between the observations, as well as the number of observations collected during the initial intervention administration period.

Using the same general setting as in Table 4, Figure 4 depicts the power to detect a carryover effect as a function of the effect size of the carryover effect (i.e., the difference in mean values in standard deviation units between an initial and subsequent intervention administration periods). A 4p × 2i design was assumed with *T* = 400. Figure 4 clearly shows that serial correlation strength and effect size influence power to detect a carryover effect, except in settings where only a few observations were obtained in the initial intervention administration period. Obviously, therefore, if one was to design an N-of-1 study accommodating tests of carryover effects by contrasting measures obtained during an initial intervention administration period with those obtained during subsequent periods and not employing washout periods, then collecting a larger number of observations during the initial administration period would be ideal. Alternatively, one could randomize the order of intervention administrations and then assess carryover effects to facilitate interpretation and reduce possible confounding.

## 8. Discussion

As interest in ‘personalized,’ ‘individualized,’ and ‘precision’ medicine grows, so will the need to prove that the interventions being recommended based on a patient’s unique biochemical and physiological profile actually work [1,13]. Unfortunately, proving that an intervention tailored to an individual patient actually works for that patient will require the use of sophisticated single subject or N-of-1 study designs and intervention response measurement devices [2,14]. We have shown how factors such as serial correlation, the manner and number of times in which different interventions are administered to a patient, heteroscedasticity, and carryover effects can either influence the power of an N-of-1 study or be modeled within the context of an N-of-1 study.

Despite the fact that our analyses and results have important implications in the design and implementation of N-of-1 studies, they do have limitations. First, we only considered the use of a linear model with an AR(1) serial correlation structure. Such models may not be applicable to all N-of-1 studies. It would be of value to assess the influence of different serial correlation assumptions on the power of N-of-1 clinical trials, as well as the robustness of inferences drawn when incorrect assumptions about the serial correlation are made, as considered in a recent set of simulation studies [9]. Second, we considered only fixed intervention periods with a pre-specified number of response collections over the course of a trial comparing two interventions. Randomization, if feasible, of the order in which interventions are administered could help alleviate confounding and carryover effects, and consideration of the minimum number of response collections needed to achieve a level of power could shed light on the efficiency of a trial. Third, we did not consider the effect of missing data, covariates and measured confounding variables, and the use of unequal numbers of observations in different intervention periods, all of which may have an impact on the power and feasibility of N-of-1 trials.

There are some obvious extensions to our studies that go beyond those needed to address their clear limitations. For example, sequential and adaptive designs might be of inherent interest and more appropriate to consider in certain applications (e.g., in the evaluation of interventions for an acute, life-threatening condition in which fixed time-length intervention administration periods for a potentially inferior treatment could do irreparable harm) [5]. In addition, designs that consider multivariate responses deserve attention since the symptoms or clinical measures one needs to assess intervention responsiveness may be multifaceted and not hinge on a single measure. Despite these issues, it is hoped that our results will not only motivate further studies but help move the adoption of objectively-determined, safe and effective tailored interventions for individual patients forward.

## 9. Conclusions

We studied the effects of various aspects of N-of-1 designs on the power to detect treatment and carryover effects. These studies considered the manner in which two interventions were administered in sequence, the strength of the serial correlation between the observations, heteroscedasticity, and the use of washout period. We found that the influence of serial correlation and heteroscedasticity on power can be substantial, but can be mitigated to some degree through the use of multiple evaluation periods with and without washout periods. We also found that the detection of certain types of carry-over effects can be heavily influenced by design considerations as well. 

## Figures and Tables

**Figure 1 healthcare-07-00084-f001:**
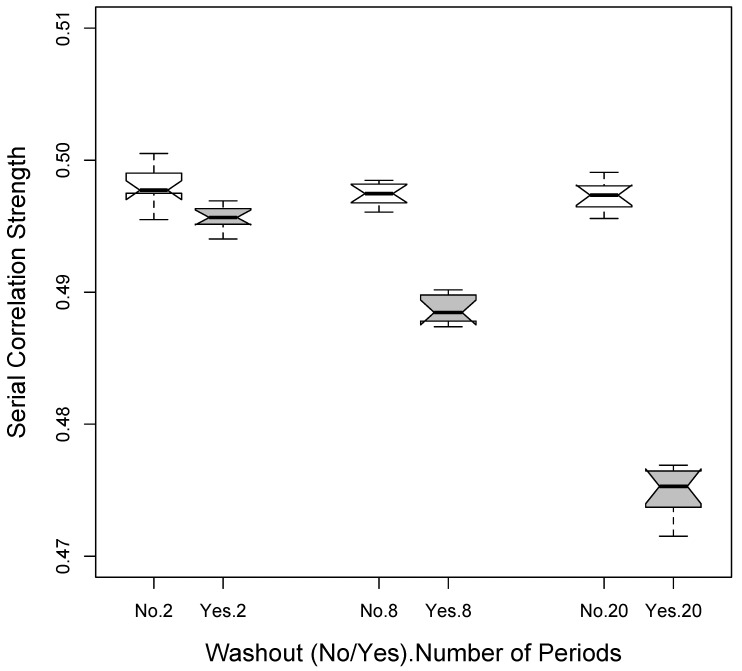
Reduction in serial correlation when washout periods are used based on simulation studies. The settings considered involved a 2 × 200 observation (2 total periods) setting, a 4 × 50 observation (8 total periods) setting and a 10 × 2 observation (20 total periods) setting.

**Figure 2 healthcare-07-00084-f002:**
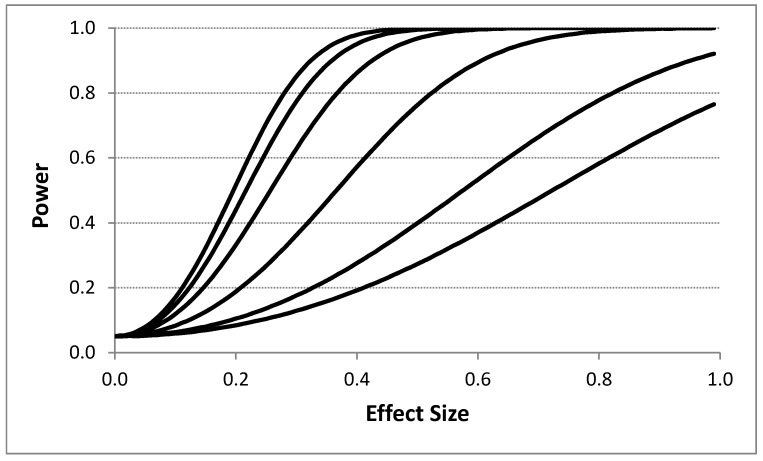
Influence of the effect size (i.e., difference between intervention responses in standard deviation units) on power assuming four intervention periods and 400 total observations (i.e., 8 intervention administration periods overall with 50 observations; a ‘4p × 2i’ design as described in the text) with no washout periods based on the proposed analytical model. Each line corresponds to a different assumed serial correlation strength. The upper most line represents a 0.0 serial correlation, with successive lines below it reflecting serial correlations of 0.1, 0.25, 0.5, 0.75, and 0.9, respectively. A type I error rate of 0.05 was assumed.

**Figure 3 healthcare-07-00084-f003:**
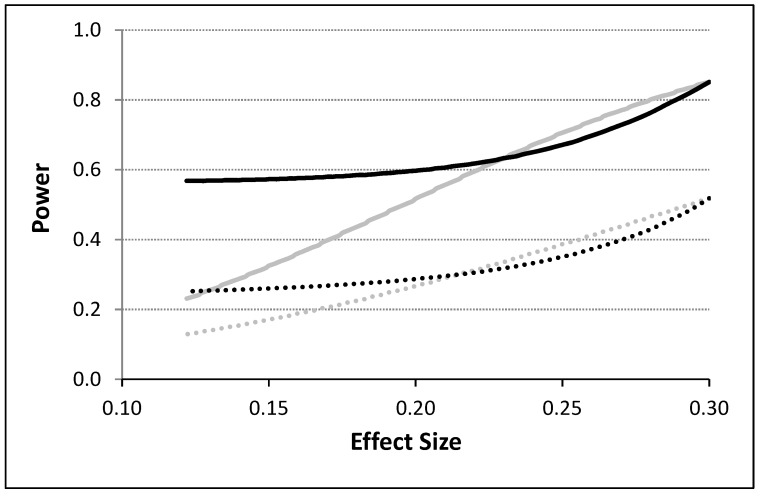
The power to detect the difference between two interventions as a function of the effect size (computed as the difference in means divided by the pooled standard deviation) for serially correlated and potentially heteroskedastic observations based on the proposed analytical model. The gray lines assume equal variances and unequal means and the black lines assume unequal variances. The dashed lines correspond to situations in which a residual serial correlation of 0.5 was assumed and the solid lines correspond to situations in which no residual correlation was assumed. A type I error rate of 0.05 was assumed.

**Figure 4 healthcare-07-00084-f004:**
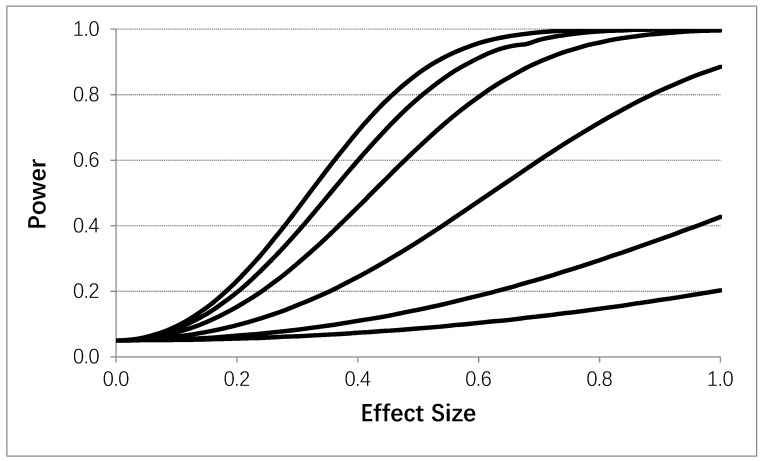
Influence of the effect size (i.e., difference between response observations obtained during a fist intervention period and subsequent intervention periods) on the power to detect a carryover effect assuming four intervention periods (i.e., 8 intervention administration periods overall) with no washout periods based on the proposed analytical model. 50 data collections were made during each intervention administration. Each line corresponds to a different assumed serial correlation strength. The upper most line representing a 0.0 correlation, with successive lines below it reflecting serial correlations of 0.1, 0.25, 0.5, 0.75, and 0.9, respectively. A type I error rate of 0.05 was assumed.

**Table 1 healthcare-07-00084-t001:** The effect of serial correlation and intervention period stacking with and without washout periods on the power to detect a mean difference of 0.30 standard deviation units between two interventions with 400 total observations based on the proposed analytical model. A type I error rate of 0.05 was assumed.

Design		Serial Correlation Strength							
Periods	Obs/Per	0.000	0.000 w	0.250	0.250 w	0.500	0.500 w	0.750	0.750 w
1p × 2i	200	0.851	0.851	0.617	0.616	0.330	0.327	0.126	0.122
2p × 2i	100	0.851	0.851	0.621	0.619	0.341	0.335	0.142	0.134
4p × 2i	50	0.851	0.851	0.628	0.624	0.362	0.351	0.176	0.159
8p × 2i	25	0.851	0.851	0.643	0.636	0.403	0.382	0.242	0.209
10p × 2i	20	0.851	0.851	0.650	0.641	0.423	0.398	0.275	0.234
20p × 2i	10	0.851	0.851	0.684	0.667	0.518	0.472	0.433	0.356
40p × 2i	5	0.851	0.851	0.745	0.716	0.674	0.602	0.681	0.569

**Key**: Periods = the number of periods in which two interventions are administered; Obs/Per = the number of observations per intervention period; w=washout period is assumed.

**Table 2 healthcare-07-00084-t002:** Simulation study results investigating the power of N-of-1 designs in comparison to the analytical results reflected in Table 1.

Setting		Serial Correlation Strength							
Periods	Effect Size	0.00 s	0.00 a	0.25 s	0.25 a	0.5 s	0.5 a	0.75 s	0.75 a
1p × 2i	0.0	0.059	0.050	0.049	0.050	0.061	0.050	0.051	0.050
4p × 2i	0.0	0.060	0.050	0.044	0.050	0.048	0.050	0.066	0.050
10p × 2i	0.0	0.061	0.050	0.063	0.050	0.051	0.050	0.060	0.050
1p × 2i	0.3	0.832	0.851	0.619	0.617	0.317	0.330	0.136	0.126
4p × 2i	0.3	0.856	0.851	0.614	0.628	0.336	0.362	0.201	0.176
10p × 2i	0.3	0.862	0.851	0.656	0.650	0.425	0.423	0.288	0.275

**Key**: Periods = the number of periods in which two interventions are administered; An ‘s’ in the column headings indicates power based on simulation studies and an ‘a’ indicates power based on the analytical model.

**Table 3 healthcare-07-00084-t003:** Simulation study results investigating the power of N-of-1 designs as a function of effect size (the difference between the experimental intervention and comparator intervention in standard deviation units) and serial correlation strength.

Setting		Serial Correlation Strength			
Periods	Effect Size	0.00	0.25	0.50	0.75
1p × 2i	0.0	0.059	0.049	0.061	0.051
4p × 2i	0.0	0.060	0.044	0.048	0.066
10p × 2i	0.0	0.061	0.063	0.051	0.060
1p × 2i	0.3	0.832	0.619	0.317	0.136
4p × 2i	0.3	0.856	0.614	0.336	0.201
10p × 2i	0.3	0.862	0.656	0.425	0.288
1p × 2i	0.6	1.000	0.993	0.851	0.359
4p × 2i	0.6	1.000	0.992	0.905	0.528
10p × 2i	0.6	1.000	0.997	0.945	0.786
1p × 2i	0.9	1.000	1.000	0.997	0.661
4p × 2i	0.9	1.000	1.000	0.994	0.842
10p × 2i	0.9	1.000	1.000	0.999	0.990

**Key**: Periods = the number of periods in which two interventions are administered.

**Table 4 healthcare-07-00084-t004:** The effect of serial correlation and intervention period stacking on the power to detect a carryover effect of 0.30 standard deviation units in an N-of-1 crossover trail with 400 total observations based on the proposed analytical model. A type I error rate of 0.05 was assumed.

Design				Serial Correlation Strength			
Periods	Obs/Per	Baseline *n*	Repeat *n*	0.000	0.250	0.500	0.750
1p × 2i	200	200	0	0.050	0.050	0.050	0.050
2p × 2i	100	100	100	0.564	0.359	0.190	0.089
4p × 2i	50	50	150	0.451	0.285	0.158	0.083
8p × 2i	25	25	175	0.289	0.188	0.116	0.074
10p × 2i	20	20	180	0.247	0.164	0.105	0.071
20p × 2i	10	10	190	0.152	0.111	0.083	0.066
40p × 2i	5	5	195	0.102	0.083	0.070	0.063

Key: Periods = the number of intervention periods in which the two treatments are compared; Obs/Per = the number of response observations per intervention period; Baseline n = the number of time points for which response data have been collected during a baseline period; Repeat n = the total number of observations for which response data have been collected beyond the initial intervention administration period.

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
