# Peer review of "Power and Design Issues in Crossover-Based N-Of-1 Clinical Trials with Fixed Data Collection Periods"

_healthcare, 2019, doi:10.3390/healthcare7030084_

Round 1

Reviewer 1 Report

Concerning the interesting topic of personalized medicine, I would suggest to discuss the "liquid dynamic medicine" concept proposed in N-of-1 trial. (Silvestris et al. "Liquid dynamic medicine and N-of-1 clinical trials: a change of perspective in oncology research." Journal of Experimental & Clinical Cancer Research 36.1 (2017): 128.

I suggested the evaluation of that publication according to its clinical relevance.
In my opinion, the paper should focused on the clinical relevance of N1-trials in the design of clinical trials evaluating targeted therapies

Author Response

Reviewer 1

Concerning the interesting topic of personalized medicine, I would suggest to discuss the "liquid dynamic medicine" concept proposed in N-of-1 trial. (Silvestris et al. "Liquid dynamic medicine and N-of-1 clinical trials: a change of perspective in oncology research." Journal of Experimental & Clinical Cancer Research 36.1 (2017): 128. I suggested the evaluation of that publication according to its clinical relevance.
In my opinion, the paper should focused on the clinical relevance of N1-trials in the design of clinical trials evaluating targeted therapies

Response: We were not aware of the paper by Silvestris et al. but have now referenced it in our paper. We agree with many points raised in the paper. For example, we agree that N-of-1 studies should be pursued in a way that makes their results clinically relevant if they are to advance personalized medicine. We emphasize this now in the paper. We also note that some N-of-1 trial designs in oncology can be complicated given the nature of the disease (e.g., cross over trials when the patient is responding well to one of the treatments and crossing them over could compromise their health) and we also point this out in the paper. We also added a few additional references for perspective.

Reviewer 2 Report

After a concise but clear introduction to single-case experimental designs and their value and challenges in medical research, three issues are tackled: the effect of serial correlation and the length of an intervention period on the statistical power (when using a linear regression model including a autoregressive parameter), the effect of heteroscedasticity on power, and the power of detecting carry-over effects.

My major remark is that the methodology is insufficiently described. There is no methods section. Instead, the simulation design is partially described under the results section (lines 169-200), and also in other places in the results section pieces of information are given. My suggestions are the following:

·       Regarding the simulation study, I would like to see a systematic description of the models used to generate the data, as well as a clear overview of what parameters were varied and what levels they take. I would also clarify the designs mentioned on line 181-182. For instance, I suppose that all ‘phases’ are equally long?

·       I propose to add a convincing rationale for the choices made. For instance, it is not clear to me to what extent the length of the time series and the alternation of the treatments are realistic in biomedical research (I doubt that this is the case, but maybe the authors can give examples of domains where this is the case).

·       In addition, I would like to see a clear description of how data are analyzed. For instance, the authors describe how data were generated with washout periods, but it is not clear whether these washout periods were explicitly accounted for in the analyses. A similar comment on heteroscedasticity: two variances are used to generate the data, but is heteroscedasticity accounted for in the analysis? If not, how could this have changed the results? is the type I error rate under control when the heterogeneity is not accounted for? It is also not very clear how exactly the carry-over effects are looked for.

Another suggestion is to repeat the structure of the methods section in the results section. For instance, only when Table 2 is discussed, it is mentioned that the results given in Table 1 were analytical results, not the results of the simulation study.

Next, I find the use of the term 2 x 2 design to refer to what (at least in the behavioral sciences) is called an ABAB reversal design is misleading: The term 2 x 2 design is commonly used to refer to a two-factorial design (with two factors, each with two levels).   

Finally, the AB-phase design is described without warning the reader that the internal validity of this design is rather low, and therefore has to be avoided.

Further, the numbering of tables and figures has to be revised (Figure 2 comes before Figure 1; there are two tables called Table 2).

Details:

line 99: drop ‘a’

line 118: add ‘to’

line 132: the error term is not assumed to follow a standard normal distribution, but rather a normal distribution with variances sigma².

line 143: ‘if’ instead of ‘it’

line 166: ‘type I error rate of’ instead of ‘type I error or’

line 216: ‘considered’ instead of ‘consider’

Author Response

Reviewer 2

After a concise but clear introduction to single-case experimental designs and their value and challenges in medical research, three issues are tackled: the effect of serial correlation and the length of an intervention period on the statistical power (when using a linear regression model including a autoregressive parameter), the effect of heteroscedasticity on power, and the power of detecting carry-over effects.

My major remark is that the methodology is insufficiently described. There is no methods section. Instead, the simulation design is partially described under the results section (lines 169-200), and also in other places in the results section pieces of information are given. My suggestions are the following:

·       Regarding the simulation study, I would like to see a systematic description of the models used to generate the data, as well as a clear overview of what parameters were varied and what levels they take. I would also clarify the designs mentioned on line 181-182. For instance, I suppose that all ‘phases’ are equally long?

·       I propose to add a convincing rationale for the choices made. For instance, it is not clear to me to what extent the length of the time series and the alternation of the treatments are realistic in biomedical research (I doubt that this is the case, but maybe the authors can give examples of domains where this is the case).

Response: We agree that improvements in the description of the motivation and technical details of our studies could be made. We also agree that a broader discussion of the practical merits of the different study designs we discussed is called for. As a result, we have changed the organization of the paper to accommodate these items. We now have sections focusing on basic design issues, practical concerns with the settings we explore, the analytical models used and the simulation studies we performed.

·       In addition, I would like to see a clear description of how data are analyzed. For instance, the authors describe how data were generated with washout periods, but it is not clear whether these washout periods were explicitly accounted for in the analyses. A similar comment on heteroscedasticity: two variances are used to generate the data, but is heteroscedasticity accounted for in the analysis? If not, how could this have changed the results? is the type I error rate under control when the heterogeneity is not accounted for? It is also not very clear how exactly the carry-over effects are looked for.

Response: We have added material that discusses how our analyses accommodated the incorporation of washout periods as well as heteroscedasticity. It is true that we generated settings in which heteroscedasticity is assumed to make the settings more realistic and to assess how robust the proposed analytical methods are, but more information on how one could accommodate or model such effects is needed and now provided.

Another suggestion is to repeat the structure of the methods section in the results section. For instance, only when Table 2 is discussed, it is mentioned that the results given in Table 1 were analytical results, not the results of the simulation study.

Response: We have re-organized the text along these lines and believe the presentation is much clearer as a result.

Next, I find the use of the term 2 x 2 design to refer to what (at least in the behavioral sciences) is called an ABAB reversal design is misleading: The term 2 x 2 design is commonly used to refer to a two-factorial design (with two factors, each with two levels).   

Response: We agree and have pointed this out in the text. We also made changes to the way we refer to the different designs we studied to avoid confusion with traditional 2 x 2 factorial designs.

Finally, the AB-phase design is described without warning the reader that the internal validity of this design is rather low, and therefore has to be avoided.

Response: We now emphasize that some designs are not known to perform well in practice, but also emphasize that this fact is the motivation for comparisons of the type we consider.

Further, the numbering of tables and figures has to be revised (Figure 2 comes before Figure 1; there are two tables called Table 2).

Response: We have made these changes.

Details:

line 99: drop ‘a’

line 118: add ‘to’

line 132: the error term is not assumed to follow a standard normal distribution, but rather a normal distribution with variances sigma².

line 143: ‘if’ instead of ‘it’

line 166: ‘type I error rate of’ instead of ‘type I error or’

line 216: ‘considered’ instead of ‘consider’

RESPONSE: We have made these changes and thank the reviewer for pointing them out.

Round 2

Reviewer 2 Report

Yes, the manuscript has improved and I think it is almost ready for being accepted. I am however not satisfied with the reply to one of my comments: "Finally, the AB-phase design is described without warning the reader that the internal validity of this design is rather low, and therefore has to be avoided." "Response: We now emphasize that some designs are not known to perform well in practice, but also emphasize that this fact is the motivation for comparisons of the type we consider." An issue with an AB-phase design is that it is possible that due to an external event around the time that the baseline phase (A) stopped and the treatment phase (B) starts, scores might change. This change can confound the treatment effect. This is especially a risk in N=1 designs, because in group designs event effects might be averaged out. This is a reason for using ABA, ABAB or multiple baseline designs: in these designs we can observe the change at multiple time points. I am not satisfied with the response of the reviewers, because this problem cannot come out of the simulation study, because they did not simulate external event effects...